# Robust Multi-task Modeling for Bayesian Optimization via In-Context Learning

## Abstract

Bayesian optimization is a sample-efficient optimization technique for black-box optimization, and leveraging historical information from related tasks can greatly improve its performance. Gaussian processes (GPs) are commonly used to model this multi-task data; however, they trade off complexity with expressivity. Jointly modeling all tasks can be computationally infeasible for GPs, while scalable approaches may fail to effectively utilize inter-task relationships. Moreover, these methods are often prone to negative transfer, where the inclusion of unrelated tasks degrades predictive performance. In this paper, we present Multi-Task Prior-Data Fitted Networks (MTPFNs), a multi-task model that efficiently and jointly models all tasks and data points. We show that MTPFNs serve as a compelling surrogate model that is robust to negative transfer, and their flexibility enables more efficient exploration. We demonstrate the effectiveness of our approach across a variety of synthetic and real-world benchmarks including hyperparameter optimization.

## 1 Introduction

Black-box optimization is widely used for tuning parameters in scientific settings and industrial applications to optimize the outputs of resource-intensive processes that do not have a known analytical form and for which gradients are not available. For example, a practitioner may wish to tune the hyperparameters of a machine learning model to maximize the log-likelihood on a validation set in AutoML [29], a chemist may aim to design a reaction by choosing the concentrations and experiment conditions to maximize the resulting product [28], or an engineer may seek to find an optimal design for a new automobile that maximizes a safety-fuel economy trade-off [21].

For these resource-intensive settings, Bayesian optimization (BO) is a sample-efficient method that aims to find the global optimum with minimal function evaluations by using probabilistic surrogate models to select future evaluations. Transfer learning can be used to further improve sample efficiency by extracting information from related tasks. For example, when tuning the hyperparameters for a machine learning model, one may have access to previous model evaluations from similar datasets or architectures. Existing transfer-learning BO approaches often use a surrogate to jointly model the data from different tasks. However, such approaches have limitations. The commonly-used Gaussian process (GP) surrogate models often trade off with data efficiency with robustness: the models which jointly fit all of the data [4] may make strong assumptions about how different tasks are correlated and experience *negative transfer*, where unrelated tasks hinder performance; meanwhile, other approaches which fit separate GPs per task and ensemble their predictions [31] are more robust to negative transfer but cannot jointly capture cross-task information. The appropriate model that strikes the right balance between efficient, flexible transfer and robustness to negative transfer is hard to compute in the classical Gaussian process setup.

Prior-data Fitted Networks (PFNs) [22] offer an attractive alternative to Gaussian Processes because they are capable of approximating the posterior for any prior over functions that can be sampled

from. This enables them to mimic the behavior of Gaussian processes, or do Bayesian inference over complex bespoke priors. However, PFNs have only been applied to single-task settings.

In this work, we propose Multi-Task Prior-data Fitted Networks (MTPFNs). By training with a novel data generating process (prior) that generates data from both related and unrelated tasks, we build MTPFNs that generalize better in classical transfer tasks compared to previous GP-based approaches, while being more robust to negative transfer from irrelevant auxiliary sources. We demonstrate the efficacy of MTPFNs as a surrogate model in Bayesian optimization on synthetic problems and real-world AutoML benchmarks.

## 2 Background and Related Work

### 2.1 Bayesian Optimization

Bayesian optimization (BO) [12] is a sample-efficient black-box optimization method. Given a function $f$ and a compact search space $\mathcal{X} \subset \mathbb{R}^D$, BO aims to find the global maximum[1] $x^* = \arg\max_{x \in \mathcal{X}} f(x)$ by iteratively selecting points to evaluate, conditional on observations made previously. Typically, the observations are corrupted by noise $y = f(x) + \epsilon(x)$ where $\epsilon(x)$ is a noise process. At each step $t$, a probabilistic model (often a Gaussian process [27]) parameterized by $\theta$ is fit to the data collected so far $\mathcal{D}_t = \{(x_i, y_i)\}_{i=1}^t$. An acquisition function $a(x; \theta)$ (e.g. expected improvement [17]), which balances exploration and exploitation, then uses the model's posterior distribution $p(f|\mathcal{D}_t, \theta)$ to determine the next point to evaluate $x_{t+1} = \arg\max_x a(x; \theta)$. Finally, we evaluate the function for the selected point $x_{t+1}$, add the new observation to the set of function evaluations $\mathcal{D}_{t+1} = \mathcal{D}_t \cup \{(x_{t+1}, y_{t+1})\}$, and proceed to the next iteration. The surrogate model is the key determinant of how successful BO will be, and therefore improvements in the model typically lead to improvements in performance [16].

In the multi-task setting, we have access to auxiliary data from functions which may be similar to the black-box function we wish to optimize. Formally, we aim to find the global optimum of a target task (denoted by task ID 0) $x^* = \arg\max_{x \in \mathcal{X}} f_0(x)$ while having access to evaluations from auxiliary tasks $\{f_k\}_{k=1}^K$, which we assumed are defined over the same search space. Compared to single-task BO, each observation consists of an additional task index $k$, and the dataset is $\mathcal{D}_t = \{(x_i, y_i, k_i)\}_{i=1}^{n_t}$. The acquisition function $a(x; \theta)$ is then used to select the next point for the target task 0. Successful multi-task BO requires a surrogate model that is able to correctly infer the relationship between the tasks and use this information effectively.

### 2.2 Multi-Task Surrogate Models

One approach to model multi-task data for BO is to jointly model the full collection of target and auxiliary data using a single GP with a multi-task kernel [4, 30, 35, 26, 18]. We will refer to these models as multi-task GPs (MTGPs). While these joint modeling approaches are effective in the low-data regime, they become computationally infeasible as we scale the number of tasks and data points due to their cubic complexity in the total number of data evaluations.

The intrinsic coregionalization model ICM, 14 is a common choice of MTGP due to its simplicity and was proposed in Swersky et al. [30] for multi-task BO. The ICM models the functions with a kernel that decomposes into two components

$$k((x, t), (x', t')) = k_{\text{inputs}}(x, x') \cdot k_{\text{tasks}}(t, t'),$$

where $k_{\text{inputs}}$ is a kernel (often RBF or Matérn) that represents the covariance between inputs and $k_{\text{tasks}}$ captures the covariance between tasks. Because this model assumes a single shared latent function across all tasks, it can efficiently transfer knowledge between similar tasks; however, the ICM model may perform poorly when this assumption does not hold and the tasks have distinct characteristics, e.g., when they should be modelled with different lengthscales.

There are other multi-task models which require weaker assumptions. For example, the linear model of coregionalization LMC, 14 is a generalization of the ICM, which uses multiple latent functions that are linearly combined for each task, instead of assuming a single latent function for all tasks. However, although this model is more flexible than the ICM, it has increased computational complexity and may lead to overfitting.

---

[1]Or equivalently the global minimum. Without loss of generality we consider maximization.

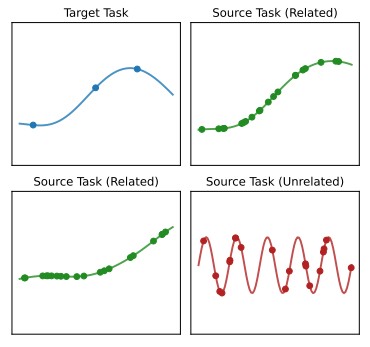
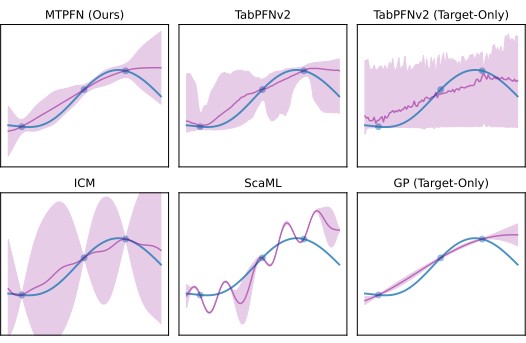

(a) Four-task problem setting  (b) Model predictions on the target task

Figure 1: **MTPFNs effectively transfer information from related tasks while remaining robust to unrelated tasks.** Compared to joint models such as ICM, ScaML, and TabPFNv2 with a categorical task variable, our MTPFN demonstrates improved robustness to the unrelated source task (red). The MTPFN is also able to borrow strength from the related source tasks (green) and outperforms models which only consider the target task. We plot the mean and 95% confidence intervals for each model.

Alternative approaches which focus on scaling to large multi-task datasets have also been proposed. Many methods fit separate GPs to each auxiliary task and ensemble their predictions to inform the target task [13, 10, 33, 7]. Although these approaches are scalable, they are not able to jointly capture information across the related tasks and instead rely on heuristics to determine the relevance of each GP. Other methods use the auxiliary data to learn a better prior over GP hyperparameters for the target task [32, 9]; however, these methods are unable to utilize the specific information within each task. Tighineanu et al. [31] propose a scalable joint modeling approach between the target task and auxiliary tasks; however, this does not model the correlations between the auxiliary tasks, thereby not taking advantage of the entire dataset. In contrast, we propose a scalable method that jointly models the full interaction between all data points and tasks.

## 2.3 Bayesian Optimization with Transformers

For single-task Bayesian optimization, there has been a growing interest in using neural-network based approaches. OptFormer [6] leverages a transformer directly trained on data collected from BO loops. In this setup the model does not only model $y$, like in classical BO, but directly predicts proposals for the next $x$. This method requires access to large amounts of domain data during training, which may not be possible in many settings.

Transformer neural processes (TNP) [24] and prior fitted networks (PFNs) [22] are transformers trained to approximate the posterior predictive distribution given a data generating function (prior). These models approximate posterior predictive distribution for a prior specified over a hypothesis space $\mathcal{H}$ where each hypothesis $h \in \mathcal{H}$ defines a relationship between inputs $x$ and outputs $y$. This work builds off of the PFN framework due to PFN's strong empirical performance in Bayesian optimization tasks [23] and prediction for tabular data [15].

A PFN, denoted by $f_\theta$, takes as input a dataset $\mathcal{D}$ and test point $x_{\text{test}}$ and outputs a distribution over the target variable $p(y_{\text{test}}|x_{\text{test}}, \mathcal{D})$. To train $f_\theta$ to approximate the posterior predictive distribution, we repeatedly sample datasets by first sampling a hypothesis $h \sim p(h)$ which defines a datasets' input-output relationship, and then sampling a dataset $\mathcal{D} \sim p(\mathcal{D}|h)$. The PFN parameters $\theta$ are optimized by minimizing the negative log-likelihood on held-out test examples across datasets, expressed as $\mathcal{L}_{\text{NLL}} = \mathbb{E}_{\mathcal{D} \sim p(\mathcal{D}|h)}[-\log f_\theta(y_{\text{test}}|x_{\text{test}}, \mathcal{D}_{\text{train}})]$, where $\mathcal{D}$ is split into $\mathcal{D}_{\text{train}} \cup \{(x_{\text{test}}, y_{\text{test}})\}$.

While TNPs and PFNs have successfully applied to Bayesian optimization in the single-task setting [23, 25], there has been no prior work which explores the use of in-context transfer of related tasks to accelerate optimization. While the PFN's pre-training can already be interpreted as meta-learning, we take it a step further by training them to do Bayesian inference over several related tasks in-context; in this sense, the multi-task PFN acts as a *meta-meta-learning* model.

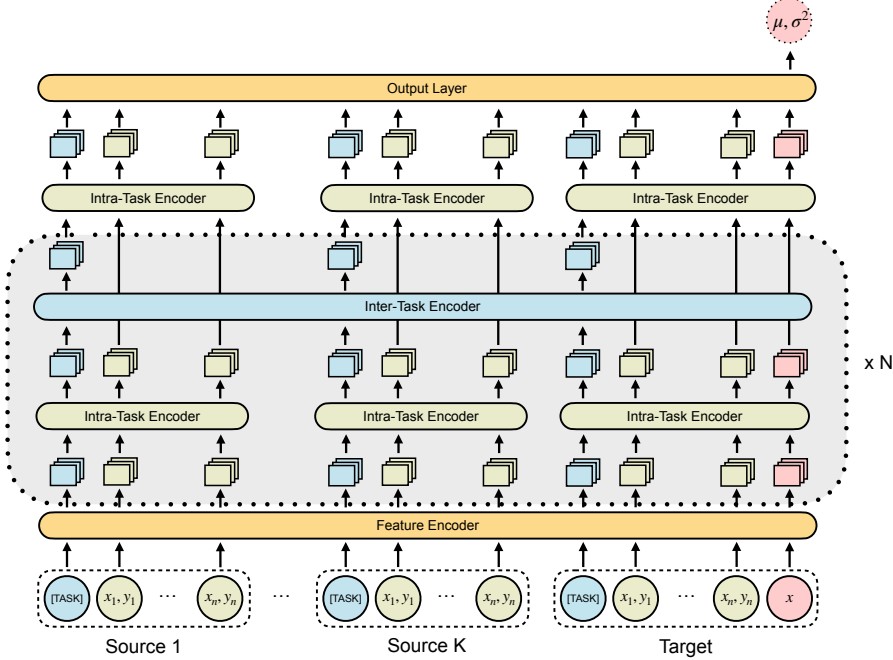

Figure 2: MTPFNs use hierarchical attention and jointly model data across information sources.

## 2.4  Long Contexts

Because of the increased number of tasks and data-points required for multi-task Bayesian optimization, the underlying architecture needs to support significantly longer context windows compared to the single-task setting. Various approaches have been proposed to extend the attention mechanisms in transformers to longer contexts, such as sparse attention [3, 36], hierarchical attention [34, 5], and others [19, 20]. See Zhuang et al. [37] for a survey of efficient methods.

## 3  Method

In this section, we present the Multi-Task Prior-Data Fitted Network (MTPFN), a scalable model that transfers relevant knowledge from auxiliary tasks to model a target task via in-context learning.

### 3.1  Data Generation Process

PFNs are trained to approximate the posterior of a data generation process (DGP), and the design of this prior has a significant influence on the model's predictive performance. While various DGPs have been proposed in previous works [e.g. 1, 15, 23], the multi-task setting which poses unique challenges. For one, multi-task problems can exhibit complex relationships between tasks, where information from one task may inform the predictions of another through shared latent structures. Additionally, real-world scenarios frequently contain unrelated or noisy tasks which do not provide any useful information; in these settings, it is important for the model to not be negatively impacted.

To address these challenges, we propose a multi-task DGP that enables the model to learn complex relationships between relevant tasks while mitigating corruption from irrelevant tasks. We first introduce inter-task relationships by sampling from an ICM MTGP, where the ICM's assumption of a shared lengthscale across tasks enables strong transfer when the tasks are related. Specifically, we sample an inter-task covariance matrix from an LKJ prior with a concentration of 1.0, which provides us with a diverse set of relationships between tasks, and we sample the shared RBF kernel lengthscale from a Gamma $(3, 6)$ prior following the default lengthscale prior in BoTorch v1.11 [2]. To prevent negative transfer, our DGP explicitly encodes the belief that each source task may be irrelevant to the target task by introducing a probability $p \in [0, 1]$ that the task is instead modeled independently using a separate RBF GP with its own lengthscale. In the following sections, we present results

under a simple and transparent DGP, but different priors and more sophisticated DGPs can easily be accommodated within the PFN framework. We present the full algorithm in Algorithm A.1, with additional discussion and ablations for alternative DGPs in Appendix A.

In Figure 1, we demonstrate the performance of various models in a synthetic multi-task setting, where three of the tasks share a latent structure and have the same length scale, while the fourth task does not contribute relevant information. We see that existing GP models and PFN priors, such as those used in TabPFNv2 [15], do not work well in this multi-task setting because these models will be influenced by the data from unrelated tasks. In contrast, our MTPFN, trained on our novel data generation procedure, is robust to the irrelevant tasks and accurately mirrors the true behavior. This underscores the benefit of our robust prior generation process.

## 3.2 Task Representation

For multi-task regression problems, it is important to consider how the task itself should be encoded within the model. This encoding can influence how the model integrates information from the various sources and impact its ability to learn helpful relationships and differentiate between tasks with irrelevant characteristics. In this section, we explore different task encoding strategies and propose a novel hierarchical attention mechanism which has many benefits over standard encoding methods.

**Categorical Feature**    For a data point $(\mathbf{x}_i, y_i)$, its associated task can be represented as a categorical feature $t_i \in \{1, 2, \ldots, T\}$, where $T$ is the number of distinct tasks. We can represent $t_i$ using a one-hot encoding $\mathbf{1}_{t_i} \in \{0, 1\}^T$, and the final input is formed by concatenating the original feature vector with the one-hot encoding $\mathbf{x}_i' = [\mathbf{x}_i; \mathbf{1}_{t_i}]$. Although simple, this approach does not provide the model with information related to the task itself. Furthermore, the maximum number of tasks must be specified at train-time, and the model is also unable to generalize to a larger number of tasks at test-time since the categorical feature has a fixed number of dimensions.

**Task Embedding**    Rather than directly using the one-hot encoding of the task, we can use a task encoder to map the task $t_i$ to a continuous embedding vector $\mathbf{e}_{t_i} \in \mathbb{R}^d$, where $d$ is the embedding dimension of the model. This embedding is jointly learned with the model parameters, allowing the task representation to adapt to task-specific characteristics. The original input $(\mathbf{x}_i, y_i)$ is first transformed into a feature $\mathbf{z}_i = \phi(\mathbf{x}_i, y_i)$, and then this feature is combined with the task embedding $\mathbf{z}_i' = \mathbf{z}_i + \mathbf{e}_{t_i}$. This approach integrates the task information directly into the feature space; however, the representation for the task is still learned independently of the information within each task, and the model remains unable to generalize to more tasks at test-time.

### 3.2.1 Hierarchical Attention Mechanism

To address these limitations, we propose a novel scalable attention mechanism for PFNs that effectively leverages the natural hierarchical structure of multi-task data, as shown in Figure 2. Our approach applies hierarchical attention [34] to the multi-task regression setting and uses specialized transformer blocks to separately model intra-task and inter-task relationships.

For intra-task encoding, we introduce a learnable "[Task]" token to each task that summarizes task-specific properties. The intra-task transformer blocks are responsible for learning the relationships of the data points within each task, By performing attention over these points, the intra-task block updates the embeddings for each data point and also updates the "[Task]" token with a summary embedding for the task, requiring $O(D^2)$ total compute per task. Then, the inter-task encoders, responsible for learning the relationship between tasks, attend to these summary "[Task]" embeddings, with $O(T^2)$ complexity. This hierarchical design reduces the overall attention complexity from the naive global setting of $O(D^2T^2)$ to $O(D^2T + T^2)$, enabling significantly longer contexts while still allowing for every data point to influence others. We interleave the intra-task and inter-task blocks in our architecture, although Chalkidis et al. [5] show that other topologies may also be effective

Our hierarchical attention directly addresses many of the limitations of other task encoders. First, our attention mechanism naturally handles inputs of varying lengths, allowing the model to generalize to any number of tasks. This flexibility ensures that even if the model encounters more tasks at test time than it did during training, it can still meaningfully integrate new task representations. Furthermore, our approach enables the model to dynamically learn task representations which depend on the data from the task, and its representation of each task evolves through the many layers of attention. This

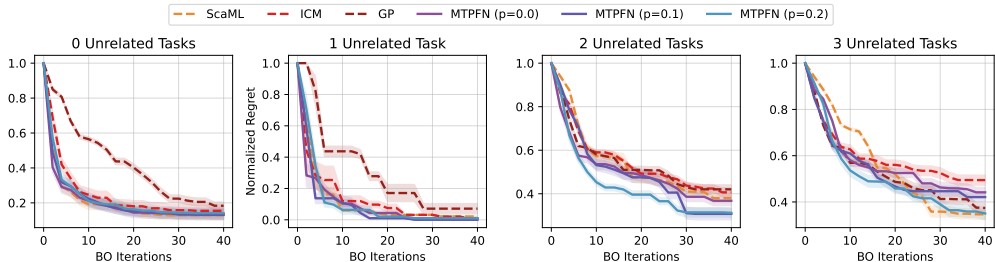

Figure 3: **MTPFNs are robust to negative transfer from unrelated tasks**. We evaluate BO across four multi-task settings, where the target task is related to {0, ..., 3} out of 3 auxiliary tasks. We compare the performance of MTPFNs with different $p$, where $p$ represents the probability that an auxiliary task was drawn independently from the target task during training. As we increase the number of unrelated tasks during evaluation, the MTPFNs which were exposed to unrelated tasks during training ($p > 0$) outperform the ICM model, which suffers from negative transfer. We plot the mean and standard error of the mean over 5 trials.

enables tasks with similar patterns to develop similar representations, allowing the model to better capture the potentially complex relationships between tasks.

## 4    Advantages of MTPFNs through empirical evaluations

MTPFNs are a compelling surrogate model for multi-task settings: they are capable of efficiently scaling to large multi-task datasets, and their flexibility enables them to effectively adapt to diverse information sources. In contrast, although multi-task GPs are commonly used for multi-task regression, these models often contain strong assumptions. Furthermore, they often trade off efficiency and expressiveness: methods which jointly model all tasks capture cross-task interactions, but are computationally expensive, while scalable methods may ignore important inter-task interactions. In this section, we provide explicit demonstrations of the strengths of MTPFNs for multi-task learning. PFNs can be trained off a wide range of data generating processes.

### 4.1    MTPFNs are robust to negative transfer

For GPs, the lengthscales are important hyperparameters that control how sensitive the covariance is to changes in the inputs. When modeling multiple tasks, it is often assumed that these tasks all share the same lengthscales (implied by the ICM model). However, this behavior may not be true in practice, and GPs with the ICM kernel may fail to accurately model the problem and suffer from negative transfer, where the inclusion of information from one task hurts the performance on another. In contrast, the flexibility of MTPFNs allow us to train them in a way that explicitly reduces the impacts of negative transfer, as explained in Section 3.1.

In Figure 3, we evaluate the performance of MTPFNs trained with varying proportions of unrelated tasks. In this evaluation setting, there are three auxiliary tasks, where one, two, or three auxiliary tasks are unrelated to the target task. When only one of the auxiliary tasks is unrelated, we find that all of the multi-task methods perform similarly. However, as we increase the number of unrelated auxiliary tasks to two out of three, we find that the MTPFNs trained on data with a higher proportion of corrupted tasks outperform the ICM, which is more sensitive to negative transfer. When we increase the number of unrelated auxiliary tasks to three out of three, we find that the MTPFN trained with $p = 0.2$ is comparable to the single-task GP, which is the underlying DGP for this problem.

### 4.2    MTPFNs efficiently model inter-task relationships

Many existing Gaussian process surrogate models trade off modeling inter-task relationships with efficiency. To demonstrate the capabilities of MTPFNs, we design a synthetic regression problem with multiple auxiliary tasks to highlight the importance of joint modeling. In this setting, all of the data points across all source tasks are drawn from the same function, and this function is highly correlated with the target task. However, there are regions of the input domain where the relevant

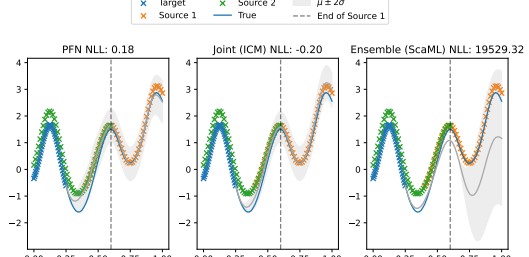
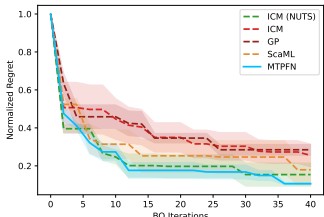

Figure 4: **MTPFNs jointly model the data from the target and all auxiliary tasks and perform fully Bayesian inference**. **(Left)**: MTPFNs perform similarly to other joint models like ICM and outperform ensemble-based models like ScaML. **(Right)**: MTPFNs have comparable performance to fully Bayesian methods like ICM with MCMC (NUTS) sampling.

source tasks do not have any overlap with the target task. Therefore, the model will only be able to make accurate predictions if it is able to leverage the relationship between source tasks.

In Figure 4 (Left), we visualize the predictive distributions of MTPFNs, ICM, and ScaML. We see it is necessary to jointly model the target task along with all of the auxiliary tasks, as done by MTPFNs and ICM, in order to accurately predict the behavior of the target task across the entire domain. In contrast, ensemble methods such as ScaML, which do not model the joint interactions between auxiliary tasks, are unable to capture the relevant information to make accurate predictions.

Although powerful, traditional joint modeling methods like ICM are unable to scale to a large number of tasks and data points. In Appendix B, we benchmark the runtimes of multi-task models as we increase the number of tasks and the number of data points per task, and we find that this problem quickly becomes unmanageable for ICM. In contrast, MTPFNs are able to scale to large amounts of data while jointly modeling all interactions.

### 4.3   MTPFNs quickly perform fully Bayesian inference

Müller et al. [22] demonstrate that transformers which are trained to minimize the negative log-likelihood over held-out data from a data-generating process naturally perform Bayesian inference by implicitly learning the posterior predictive distribution. Specifically, when the model is trained to minimize the expected NLL across sufficiently many datasets which are sampled from the data-generation prior, the final model outputs a posterior predictive distribution which marginalizes over all of the possible samples from the prior which are consistent with the observed data.

The ability for PFNs to perform fully Bayesian inference also holds in the multi-task setting: we train MTPFNs on datasets sampled from the ICM model. In Figure 4 (Right), we use an ICM to generate 5 different multi-task datasets with 3 input dimensions, each with 2 samples from the target task and 20 samples from each of the 3 auxiliary tasks. The auxiliary tasks have varying amounts of correlations with the target task. We then perform 10 runs of Bayesian inference for each multi-task dataset and summarize the results. We find that MTPFNs perform comparably to the fully Bayesian inference using MCMC sampling through NUTS. This fully Bayesian approach is particularly helpful in the setting where there are very few observations per task and thus there should be high uncertainty over the true inter-task covariance. The ICM model with MAP estimation does not account for this uncertainty and under-performs in this setting. Furthermore, the MTPFN is able to make predictions using one forward pass of the model in approximately 0.5 seconds on average, while NUTS takes orders of magnitudes longer at 352 seconds per iteration. We showcase further demonstrations of the importance of fully Bayesian inference in Appendix B.

### 4.4   MTPFNs can leverage domain data

When making predictions with PFNs, there are various methods to incorporate domain data to improve the performance. One approach is fine-tuning, where the parameters of a base model are updated to adapt to the specific characteristics of the target domain. This method enables the PFN to specialize to the particular domain; however, fine-tuning is computationally expensive and requires

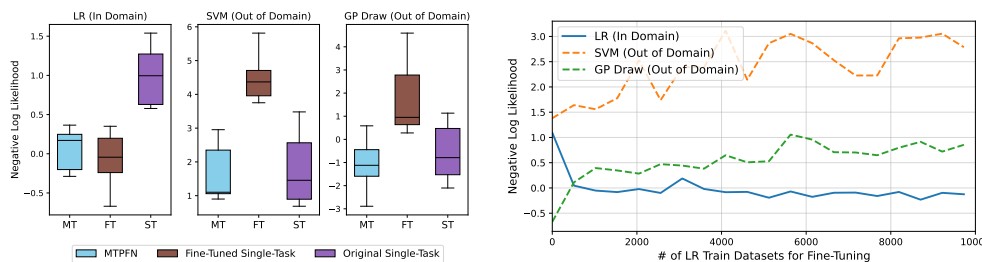

Figure 5: **MTPFNs, which use domain data through in-context learning, match the performance of Fine-Tuned PFNs on in-domain data while generalizing better to other domains.** (**Left**): MTPFNs have comparable NLLs to Fine-Tuned PFNs on in-domain data (LR) and outperform Fine-Tuned PFNs on other domains (SVM and GP Draws). (**Right**): As we fine-tune on more in-domain data, the NLL for Fine-Tuned PFNs significantly worsens for other domains.

updating the model weights. Furthermore, this method is sensitive to training hyperparameters such as the amount of data and learning rate, and it is also possible to overfit and hurt generalization.

Alternatively, these additional sources of data can be provided in an in-context manner to an MTPFN. In this setting, the models are exposed to general multi-task data during training, allowing the model to learn patterns across tasks. During inference, the model uses in-context learning to make predictions and utilizes the auxiliary information without the need for parameter updates.

We demonstrate the benefits of providing domain data through in-context learning compared to fine-tuning by comparing the two approaches on the HPOBench dataset [8] for Logistic Regression (LR), which contains 25 tasks with 4 held out for evaluation. For fine-tuning, we first train a general-purpose single-task PFN using draws from a Gaussian process with an RBF kernel (see Appendix C for training details). We then fine-tune this model on the domain data. We see in right subplot in Figure 5 that as we fine-tune on more samples from the Logistic Regression dataset, the performance of the PFN on the Logistic Regression evaluation set does improve; however, the model loses the ability to generalize to other types of datasets. Specifically, we evaluate the negative log likelihood on another domain from the HPOBench dataset that optimizing SVM parameters rather than logistic regression. We see the performance of the fine-tuned model decreases significantly in this setting, because it has overfit to LR. Similarly, we test on another domain consisting of data points generated from GP draws, and we find similar deterioration in performance.

In contrast, when we pass auxiliary information into the context of a MTPFN, we are able to achieve similar performance to the results after fine-tuning. Furthermore, the in-context approach does not deteriorate performance over other datasets such as GP draws and SVM hyperparameters. In-context learning is also computationally efficient and does not require any updates to the model parameters.

## 5 Optimization benchmarks

We demonstrate the effectiveness of MTPFNs across various transfer learning tasks for machine learning hyper-optimization. We show that the models are able to effectively utilize domain data while remaining robust to negative transfer in the context of Bayesian optimization.

For our empirical results, we use a transformer backbone with 24 attention layers, where twelve intra-task attention layers are interwoven between eleven inter-task layers. Each attention layer has 4 attention heads with a hidden size of 512. The model is trained on approximately 50 million synthetically generated datasets as described in Section 3.1, with a batch size of 16 and AdamW with a learning rate of 1e-4 and cosine annealing.

We compare our method, MTPFN, to several baselines: (1) ICM [14], a joint method which trains a multi-task GP on the combined target and auxiliary data; (2) ScaML [31], an ensemble method that fits individual GPs to each auxiliary task; and (3) a single-task GP which only uses the target task and ignores the auxiliary tasks. Our Bayesian optimization results were implemented using BoTorch [2] and GPyTorch [11], and we will provide access to our code upon acceptance.

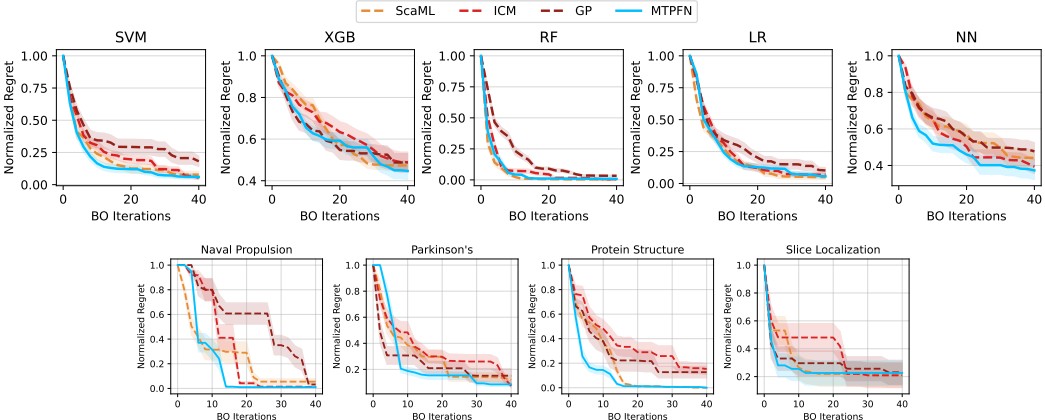

Figure 6: **MTPFNs are competitive across many hyperparameter optimization benchmarks.**
Each plot shows the normalized regret for Bayesian optimization loop that was initialized with 3
auxiliary tasks, 20 observations from auxiliary task, and 5 observations from the target task. **(Top):**
HPOBench benchmarks **(Bottom):** Tabular FC-Net benchmarks.

## 5.1 Benchmarks

We compare the effectiveness of the methods on a set of hyperparameter optimization problems for
machine learning model through HPOBench [8], a collection of tabular benchmarks with hyperpa-
rameters and their corresponding loss for models in various settings, as well as the popular FC-Net
benchmarks from Eggensperger et al. [8].

Following Tighineanu et al. [31], we consider the hyperparameter optimization for five types of
models: support vector machines (SVM), logistic regression (LR), XGBoost (XGB), neural networks
(NN), and random forest (RF). For each setting, we randomly sample one task to be the target function,
and we sample 3 auxiliary tasks from the meta-data. We randomly sample 5 points from the target
task and 20 points from each of the auxiliary tasks to use as the initialization for Bayesian inference.
We measure the normalized regret $(f^* - f_{\text{best}})/(f^* - f_0)$ where $f^*$ is the optimal value, $f_{\text{best}}$ is the
best value so far, and $f_0$ is the initial value. We run 100 replicates, each with a different combination
of target task and auxiliary task initializations, and we plot the mean and one standard error.

We share the results of our benchmark in the top panel of Figure 6, and MTPFNs are competitive
across all of the model types. Specifically, we find that in instances where the meta-tasks contain
helpful information (ScaML and ICM outperform GP), the MTPFNs are also able to effectively
utilize this data. Furthermore, in cases like XGB where there is negative transfer for the ICM model,
we find that MTPFNs are more robust and perform similarly to the standard single-task GP.

We also consider tabular FC-Net benchmarks from: Slice Localization, Protein Structure, Naval
Propulsion, and Parkinson's Telemonitoring. For each benchmark, we set the benchmark to be the
target task, and we use the three other tabular dataset as the auxiliary tasks. For instance, the results
for Slice Localization use Protein Structure, Naval Propulsion, and Parkinson's Telemonitoring as
auxiliary data sources. We initialize our Bayesian optimization problem with a random sample of 5
points from the target task and 20 points from each auxiliary task. We report the average normalized
regret over 20 trials in the bottom panel of Figure 6. MTPFNs work well on these tabular datasets,
often outperforming the other baselines.

## 6 Discussion

In this work, we present MTPFNs, a scalable and robust surrogate model for Bayesian optimization.
By jointly modeling multiple information sources through in-context learning, MTPFNs are able to
effectively use historical data. We also introduce a novel data-generation process which enables the
model to be more robust to negative transfer, and our empirical results demonstrate that our method
is competitive across a wide range of multi-task benchmarks.

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

# A  Data Generation Processes for PFNs

MTPFNs are flexible and have the capacity to incorporate various data generation processes during training. In this section, we explore the impacts of various data generation processes for MTPFNs, each designed to capture distinct inductive biases which improve model performance across different types of tasks.

## A.1  Robust Isotropic Full-Rank ICM

---

**Algorithm A.1** Data Generation Using a Robust Isotropic Full-Rank ICM

---

**Require:** Sequence length $n$, number of tasks $T$, unrelated task probability $p$
    ▷ **Input Sampling and Task Assignment**
1: Sample inputs $\{x_i\}_{i=1}^n \sim \text{Uniform}([0,1]^d)$
2: Sample task proportions $\pi \sim \text{Dirichlet}(\alpha)$
3: **for** $i = 1$ to $n$ **do**
4:     Sample task ID $t_i \sim \text{Categorical}(\pi)$
5: **end for**
    ▷ **Isotropic ICM Covariance Structure**
6: Sample task covariance matrix $K_T \sim \text{LKJ}(\eta = 1)$
7: Sample input lengthscale $\ell \sim \text{Gamma}(3, 6)$
8: Define input covariance $K_X$ on $\{x_i\}_{i=1}^n$ as RBF kernel with lengthscale $\ell$
9: Compute full ICM kernel $K = K_T \otimes K_X$
10: Sample $y \sim \mathcal{N}(0, K)$
    ▷ **Sampling Unrelated Tasks**
11: **for** each source task $j$ **do**
12:     With probability $p$:
13:         Sample new lengthscale $\ell_j \sim \text{Gamma}(3, 6)$
14:         Define RBF kernel $K_X^{(j)}$ on $\{x_i : t_i = j\}$ using $\ell_j$
15:         Resample $y^{(j)} \sim \mathcal{N}(0, K_X^{(j)})$
16: **end for**

---

In the main text, we train PFNs with the data generation process described in Algorithm A.1: we sample datapoints across tasks from a full-rank isotropic ICM model. This approach assumes that all the input dimensions share identical lengthscales; this assumption imposes a strong prior on the relationship between tasks, and enables effective information transfer across related tasks when the assumption is met. This data generation process enables us to learn information across related tasks, since the full-rank isotropic model makes strong assumptions.

To improve our model's robustness to negative transfer, we also incorporate an additional hyperparameter $p \in [0,1]$ which dictates the relatedness of the tasks during training. Specifically, $p$ is the probability that any given source task is drawn independently from the target task, and thus may have completely different behaviors and lengthscales. Our data generation procedure enables the model to see a diverse group of datasets which consist of a mix of related and unrelated source tasks.

This $p$ hyperparameter plays a crucial part in the robustness of the model against negative transfer: because the model is able to see many examples of unrelated tasks during training, it becomes more robust to seeing unrelated tasks during inference time and is less likely to be negatively impacted from irrelevant information.

In Figure A.1, we study the impact of $p$ on the model's ability to accurately predict the empirical data from HPOBench. Specifically, for each model type (SVM, LR, XGB, NN, and RF), we randomly sample one task to be the target task, and we sample 3 auxiliary tasks from the metadata. The target task is randomly initialized with 5 samples, and we also sample 20 points for each of the auxiliary tasks. We measure the mean squared error (MSE) and the negative log-likelihood (NLL) of each surrogate model on heldout examples from the target task, and we repeat this procedure 25 times and plot the average MSE and NLL for each trial.

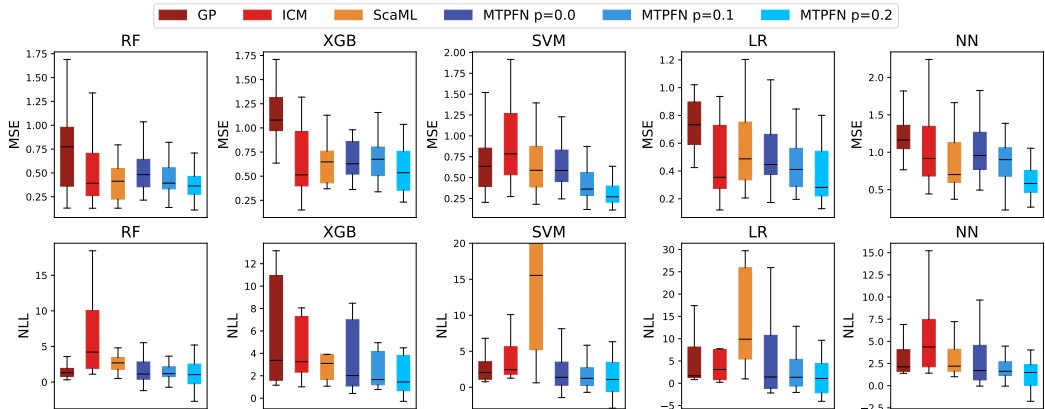

Figure A.1: As we increase $p$ (the probability that each source task is unrelated to the target task during data generation), the model becomes more robust to negative transfer and achieves better performance on real-world benchmarks. We visualize the model's predictive performance on the HPOBench dataset, where we sample 5 data points from the target task and 20 data points each from three source tasks. We plot the average MSE and NLL on holdout data from the target task across 25 trials.

We find that increasing $p$, which increases the diversity of the data that the model sees during training, leads to improved model performance on real-world benchmarks. We see that the MTPFN trained with $p = 0.2$ consistently outperforms other MTPFNs trained with lower values of $p$, and this MTPFN also outperforms baselines such as the standard ICM model, which assumes that all tasks share the same lengthscale.

## A.2 Full-Rank ICM with Automatic Relevance Determination

---

**Algorithm A.2** Data Generation Using a Robust ARD Full-Rank ICM

---

**Require:** Sequence length $n$, number of tasks $T$, unrelated task probability $p$
    ▷ **Input Sampling and Task Assignment**
1: Sample inputs $\{x_i\}_{i=1}^n \sim \text{Uniform}([0,1]^d)$
2: Sample task proportions $\pi \sim \text{Dirichlet}(\alpha)$
3: **for** $i = 1$ to $n$ **do**
4:     Sample task ID $t_i \sim \text{Categorical}(\pi)$
5: **end for**
    ▷ **ARD ICM Covariance Structure**
6: Sample task covariance matrix $K_T \sim \text{LKJ}(\eta = 1)$
7: Sample independent input lengthscales: $\boldsymbol{\ell} = (\ell_1, \ldots, \ell_d) \sim \text{Gamma}(3, 6)^d$
8: Define input covariance $K_X$ on $\{x_i\}_{i=1}^n$ as an RBF kernel with ARD lengthscales $\boldsymbol{\ell}$
9: Compute full ICM kernel $K = K_T \otimes K_X$
10: Sample $y \sim \mathcal{N}(0, K)$
    ▷ **Sampling Unrelated Tasks**
11: **for** each source task $j$ **do**
12:     With probability $p$:
13:         Sample new lengthscale $\ell_j \sim \text{Gamma}(3, 6)$
14:         Define RBF kernel $K_X^{(j)}$ on $\{x_i : t_i = j\}$ using $\ell_j$
15:         Resample $y^{(j)} \sim \mathcal{N}(0, K_X^{(j)})$
16: **end for**

---

We can also relax the assumption that all of the input dimensions share the same lengthscale, and instead sample datapoints from an ICM model with Automatic Relevance Determination (ARD), where we assume that each input dimensions has an independent lengthscale. This enables the PFNs to have more flexibility and fit more complex problems; however, this weaker assumption may reduce the model's ability to effective transfer information compared to the isotropic settings. We describe

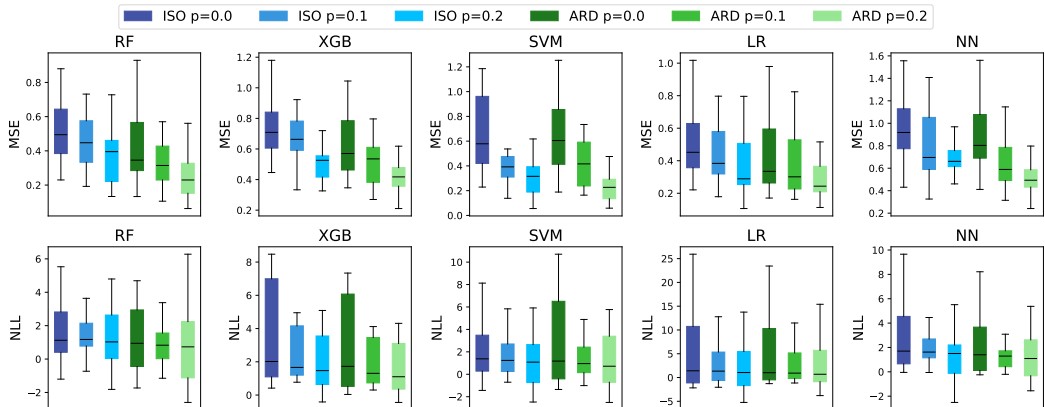

Figure A.2: MTPFNs trained with the ARD data generation process tend to outperform MTPFNs trained with the isotropic process (ISO) and achieve lower MSE and NLLs on HPOBench problems.

this data generation process in Algorithm A.2 and highlight the differences from the isotropic data generation in green.

In Figure A.2, we compare the performance of the MTPFN trained with isotropic lengthscales (ISO) to the performance of the MTPFNs trained with the ARD lengthscales. This experiment follows an identical setup to Figure A.1, where we sample 5 points from a target task and 20 points each from 3 source tasks, and evaluate the MTPFNs on held-out data from the target task. We plot the experiments across 25 trials.

We find that the improved flexibility of the ARD lengthscale generally enables the model to have better performance on the testing data, with the ARD outperforming ISO across many datasets. However, in some settings such as SVM, we find that the model performance of the isotropic ICM and the ARD ICM are comparable. This similar performance may be because the assumption of the shared lengthscale across input dimension is satisfied in this setting, so the additional flexibility of the ARD is unnecessary.

## B  Additional Empirical Results

MTPFNs are able to do Bayesian inference with a single forward pass. Furthermore, our proposed hierarchical attention mechanism enables the MTPFN to scale in $O(TD^2 + T^2)$, where $D$ is the number of data points per task and $T$ is the number of tasks. We compare the runtime of MTPFNs to joint-modeling methods such as ICM and ensemble-based methods such as ScaML in Figure A.3. We see that MTPFNs are able to perform inference on an order of magnitude more data points and tasks compared to traditional GP methods.

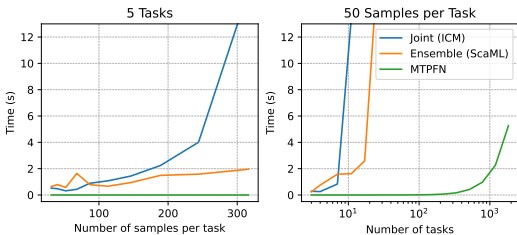

Figure A.3: MTPFNs are significantly faster than alternative GP-based methods.

When trained on a data-generation process that draws samples from a multi-task GP with an ICM kernel, we see in Figure A.4 that the MTPFN and the MTGP (ICM kernel with MAP estimation) have comparable behavior across varying levels of correlations. Furthermore, in low-data settings demonstrated by Figure A.5, we find that the MTPFN outperforms the MTGP because it considers the uncertainty over the task covariance matrix. This demonstrates that fully Bayesian inference may be preferable to MAP estimation.

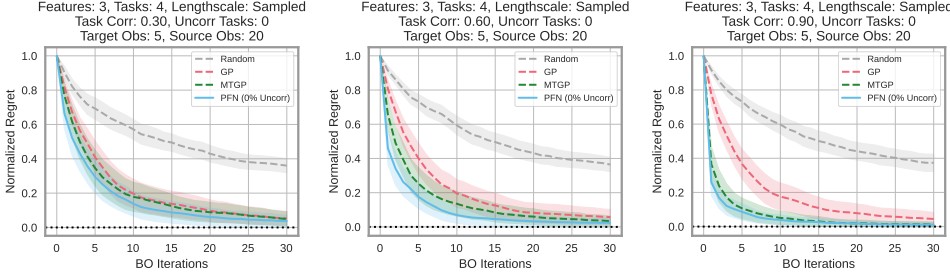

Figure A.4: ICM PFNs are comparable to MTGPs across varying levels of correlations between tasks.

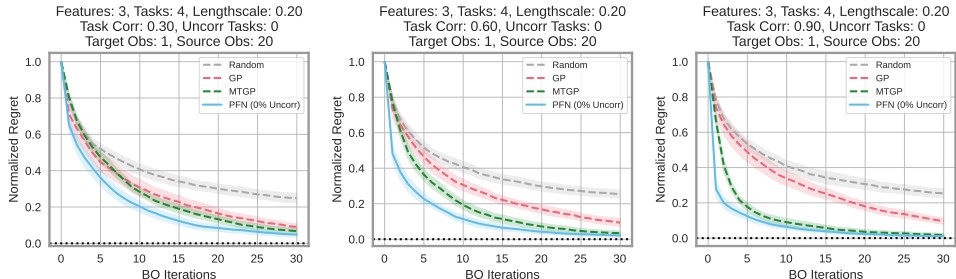

Figure A.5: In low-data settings, ICM PFNs, which approximate fully Bayesian inference, outperform MTGPs with MAP estimation. ICM PFNs are comparable to MTGPs across varying levels of correlations between tasks.

## C  Training deails

### C.1  Single-Task PFN Training

The data generation process for the single-task PFN randomly samples inputs $x$ from the unit cube, and then samples the corresponding outputs $y$ by drawing a sample from a GP with an RBF kernel with a lengthscale sampled from Gamma(3, 6). For this experiment, we use a fixed feature size of 2.

We train an 8-layer standard transformer (not hierarchical attention) with an embedding size of 256 on this data generation process for 4 million sampled datasets, with a batch size of 16, and AdamW with a learning rate of 1e-4 and cosine annealing.

### C.2  Fine-tuning Single-Task PFN

To fine-tune on the LR dataset, we develop a subsampling data-generation procedure: On the 20 training tasks, we subsample within one task to get 50 $x, y$. We uniformly select some number of them to be used as ICL training, and the remaining to be used as the test.

We fine-tune our model with a batch size of 16, and AdamW with a learning rate of 1e-4 and cosine annealing.