# OpenReview forum: "Robust Multi-task Modeling for Bayesian Optimization via In-Context Learning"
_NeurIPS.cc/2025/Workshop/Reliable_ML — NeurIPS 2025 - Reliable ML Workshop_

### Official Review · Reviewer_mxE5 · 2025-09-19
**Review of Submission 130**

**Rating:** 8
**Confidence:** 3

**Review:**

**Summary**

The paper tackles multi-task Bayesian optimization (BO) where leveraging past tasks often helps but standard Gaussian-process surrogates either don’t scale or suffer negative transfer when unrelated tasks are included. It proposes Multi-Task Prior-Data Fitted Networks (MTPFNs): PFN-based surrogates trained on a novel multi-task data-generation process that mixes related and intentionally unrelated tasks, plus a hierarchical attention architecture that captures intra-task and inter-task structure at scale. The justification is that PFNs can approximate posterior predictives for rich priors, enabling robust transfer while remaining computationally efficient. Empirically, MTPFNs show robustness to unrelated tasks and competitive or superior BO performance on synthetic settings, HPOBench, and FC-Net benchmarks.


**Strengths**

It offers a clear, scalable surrogate that jointly models many tasks while staying robust to negative transfer, and the experiments consistently tie that claim to outcomes across diverse benchmarks. The hierarchical attention + PFN training story is cohesive and easy to read.

**Weaknesses**

The narrative is dense and assumes PFN familiarity, so readers may need to work to grasp when MTPFNs are preferable to classic MTGPs and how to set the “unrelated task” probability p in practice.

**Suggestions**

Wrt the weakness, it could help to provide a practical “when/how” guide for using MTPFNs vs MTGPs, since that’s what a non-expert reader needs.

---

### Official Review · Reviewer_9pz8 · 2025-09-20
**Good paper, accept**

**Rating:** 7
**Confidence:** 2

**Review:**

## Summary

This paper presents the MTPFN method, which is a scalable and robust surrogate model for Bayesian optimization. This method jointly models multiple information sources through in-context learning. By introducing a novel data generation process and a hierarchical attention mechanism, MTPFNs achieve robustness against negative transfer from irrelevant tasks while scaling efficiently to large multi-task datasets. Compared to the baselines, this method suggests better performance and robustness against irrelevant information.

## Strengthen
1. This work proposes a new method for multi-task settings, which has not been studied in previous work.
2. Compared with baselines, this method shows better performance and Robustness.
3. The hierarchical attention mechanism shows a better complexity compared with the previous method, which ensures its scalability.

## Weakness
1. The evaluation of the model is mainly on the synthetic dataset without testing on real-world tasks. In the real task, more complexity and noisy data, and a task are expected.